

# miR-200a-3p overexpression alleviates diabetic cardiomyopathy injury in mice by regulating autophagy through the FOXO3/Mst1/Sirt3/AMPK axis

Penghua You, Haichao Chen, Wenqi Han and Jizhao Deng

Department of Cardiology, Shaanxi Provincial People's Hospital, Xi'an, China

## ABSTRACT

**Objective:** Hyperglycemia and insulin resistance or deficiency are characteristic features of diabetes. Diabetes is accompanied by cardiomyocyte hypertrophy, fibrosis and ventricular remodeling, and eventually heart failure. In this study, we established a diabetic cardiomyopathy (DCM) mouse model to explore the role and mechanism of miR-200a-3p in DCM.

**Methods:** We used db/db mice to simulate the animal model of DCM and the expression of miR-200a-3p was then examined by RT-qPCR. Tail vein injection of mice was done with rAAV-miR-200a-3p for 8 weeks, and cardiac function was assessed by cardiac ultrasound. The levels of myocardial tissue injury, fibrosis, inflammation, apoptosis and autophagy in mice were detected by histological staining, TUNEL and other molecular biological experiments.

**Results:** miR-200a-3p expression levels were significantly decreased in the myocardium of DCM mice. Diabetic mice developed cardiac dysfunction and presented pathological changes such as myocardial injury, myocardial interstitial fibrosis, cardiomyocyte apoptosis, autophagy, and inflammation. Overexpression of miR-200a-3p expression significantly ameliorated diabetes induced-cardiac dysfunction and myocardial injury, myocardial interstitial fibrosis, cardiomyocyte apoptosis, and inflammation, and enhanced autophagy. Mechanistically, miR-200a-3p interacted with FOXO3 to promote Mst1 expression and reduce Sirt3 and p-AMPK expression.

**Conclusion:** In type 2 diabetes, increased miR-200a-3p expression enhanced autophagy and participated in the pathogenic process of cardiomyopathy throug7 Mst1/Sirt3/AMPK axis regulation by its target gene FOXO3. This conclusion provides clues for the search of new gene targeted therapeutic approaches for diabetic cardiomyopathy.

## INTRODUCTION

As a metabolism related disease, which can lead to multi organ, multi system damage and dysfunction, including the cardiovascular system, diabetes has become a global health issue. As it is widely known, cardiovascular complications are considered a major cause of

Corresponding author
Jizhao Deng,
Dengdazhao1987@163.com

morbidity and mortality in patients with diabetes (*Al Hroob et al., 2019*). Diabetic cardiomyopathy (DCM) is an essential etiology of heart failure-related rehospitalization and death. First described by *Rubler et al. (1972)*, DCM, defined as myocardial dysfunction occurring independently of coronary artery disease, hypertension, or valvular heart disease among diabetic patients. Clinically, approximately 30–60% of diabetes cases are diagnosed with DCM (*Rydén et al., 2013*). Multiple biological processes have been used to explain the pathogenesis of DCM, including but not limited to oxidative stress, abnormal glycolipid metabolism, insulin resistance, calcium ion imbalance, excessive endoplasmic reticulum stress, endothelial dysfunction, abnormal mitochondrial function, and autophagy (*Delbridge et al., 2017*; *Liu, Wang & Cai, 2014*), but research into the pathogenesis of DCM has not yet translated into an effective weapon for combating DCM. At present, DCM treatment is mainly directed at diminishing blood glucose and together with the administration of anti-heart failure treatment. Therefore, finding key intervention targets is of great significance for the prevention and treatment of DCM.

MicroRNA (miRNA, miR), widely present in living organisms, is a class of non-coding single stranded RNA with 22–23 bases in length. Many authoritative literatures have shown that miRNAs are involved in various life processes of the organism and play important roles in tissue development, cell differentiation, proliferation, metabolism, apoptosis, autophagy and other biological behaviors (*Yekta, Tabin & Bartel, 2008*; *Bartel & Chen, 2004*). Meanwhile, abnormalities in the function of miRNAs are involved in various disease processes. Recent studies have shown that miRNAs, especially those expressed in abnormal abundance in the heart, play an important role in the pathogenesis of various heart diseases (*Ikeda et al., 2007*; *Thum et al., 2007*). *In vitro*, researchers have studied and regulated the expression of miRNA in cardiomyocytes, which can lead to hypertrophy or volume reduction of cardiomyocytes, revealing the biological role of miRNA in the pathological processes such as myocardial hypertrophy and ventricular remodeling (*Gao et al., 2019*; *Sygitowicz, Maciejak-Jastrzębska & Sitkiewicz, 2020*). As research has progressed and microRNAs have been found to be involved in regulating the process of myocardial injury in insulin resistance, their role in DCM has become increasingly clear. miR-320 is one of the etiological factors inducing DCM, which exacerbates cardiac lipotoxicity by interacting with CD63, thereby altering the cardiac phenotype (*Li et al., 2019*). It was found that upregulated expression of miR-30c contributed to the alleviation of DCM-related cardiomyocyte apoptosis and cardiac dysfunction (*Yin et al., 2019*). Studies from recent years also showed that miR-200a-3p expression was significantly decreased in cardiomyocytes in microembolization-induced myocardial injury model (*Chen et al., 2021*), suggesting an important role for miR-200a-3p in cardiovascular disease. Recently in the field of tumors, decreased miR-200a-3p expression within cancer tissues and induced tumor growth and development (*Wu et al., 2022*; *Shi et al., 2019*; *Li et al., 2022*). In necrotizing enterocolitis, *Liu et al. (2021)*, also observed that miR-200a-3p was lower expressed in serum of patients, which were associated with inflammation and necrosis in intestinal epithelial cells. However, as of now, the expression profile of miR-200a-3p in DCM has not been well defined.

The present study raised the hypothesis that miR-200a-3p may have decreased expression in a mice model of DCM. Increased miR-200a-3p expression could ameliorate cardiomyocyte apoptosis and myocardial interstitial fibrosis by enhancing autophagy, participating in cardiac dysfunction under diabetic condition. Further bioinformatics analysis was performed to explore the mechanism of miR-200a-3p regulation in the pathological process of diabetic cardiomyopathy.

## MATERIALS AND METHODS

### Animal model

Thirty 8-week-old healthy male C57BL/Ks db/db mice and ten control C57BL/Ks mice (db/+) (19–22 g) were selected as experimental animals and purchased from the experimental animal center of Xi'an Jiaotong University (SCXK (Shaan) 2020-001). All mice were housed in a temperature of 26 °C and humidity of 50–60% environment. The ten control C57BL/Ks mice (db/+) were acted as a control group. All C57BL/Ks db/db mice were randomly divided into three groups with 10 mice in each group: DCM group, DCM group given recombinant adeno-associated virus 9 (rAAV-9) incorporating scramble control (DCM+rAAV-SC) and DCM group given rAAV-9 incorporating miR-200a-3p mimic (DCM+rAAV-miR-200a-3p). The rAAV9 system was purchased from Hanbio Technology, Ltd. (Shanghai, China). miR-200a-3p mimic (5′-UAA CAC UGU CUG GUA ACG AUG U-3′) and scrambled control (NC mimic; 5′-UUG UAC UAC ACA AAA GUA CUG-3′) were obtained from GenePharma Biotechnology (Shanghai, China). The rAAV-miR-200a-3p or rAAV-SC were injected into the tail vein of mice at a dose of $1 \times 10^{11}$ vector genomes (v.g.) in 200 μL PBS per mouse at 8 weeks (*Yin et al., 2019*). The control group was fed on a regular chow diet, and the DCM group, DCM+rAAV-SC group and DCM+rAAV-miR-200a-3p group was fed on a high-fat diet (16% fat and 0.30% cholesterol) until the end of the experiment. At the age of 28 weeks, mice were subjected to echocardiography and then euthanized for further experiments. All experimental procedures and protocols were approved by the Ethics Committee of Shaanxi Provincial People's Hospital.

### Histopathology examination

Cardiac tissues were removed from paraformaldehyde solution, dehydrated by sequential immersion in 70–80–95–100% ethanol solutions, placed in xylene to be transparent, and finally processed for paraffin embedding and sectioned into 4–6 μm thick slices. After deparaffinization with xylene and graded hydration with ethanol solution, hematoxylin and eosin stained sections were added dropwise, respectively. Subsequently, the sections were rinsed through 95% ethanol, xylene transparent, and neutral gum was added dropwise. Myocardial tissue was observed under a microscope (Nikon, Tokyo, Japan) and photographed.

### Echocardiographic assessment

After 10 min, mice were immobilized in a supine position on an operating table. Using a VIVID7 cardiac color ultrasound diagnostic instrument (GE, Boston, MA, USA), the

probe was placed perpendicularly on the left chest wall of mice, and the left ventricular end systolic and end diastolic internal diameters (LVEDD, LVESD) were read by M-mode sampling line on the horizontal short axis view of the papillary muscle of the left ventricle, and the left ventricular ejection fraction (LVEF) and left ventricular fractional shortening (LVFS) were calculated. The numerical values of three consecutive cardiac cycles were averaged for each index, and a specific person was arranged for the examination of ultrasound and data analysis.

## Myocardial enzyme determination

After anesthetizing mice in each group with 5% isoflurane and maintaining with 1.5% isoflurane, carotid blood was quickly collected from mice into 2 ml EP tubes, and after 8 h of rest at 4 °C, cells were centrifuged at 3,000 r/min for 15 min, and the supernatant was collected as serum. Serum was analyzed for the expression of phosphocreatine kinase (CK), creatine kinase MB (CK-MB), and lactate dehydrogenase (LDH) using a semiautomatic biochemical analyzer.

## Masson's trichrome staining

After mice were anesthetized with i.p. ketamine (80 mg/kg)/xylazine (8 mg/kg) and sacrificed, the hearts were quickly excised, rinsed in cold PBS buffer, fixed in 40 µg/L paraformaldehyde for 72 h, dehydrated through ethanol, xylene, and cut into 5 µM thick sections. Nuclei were stained using wiegert's iron haematoxylin solution for 5 min. After sections were rinsed three times with distilled water, 0.7% Masson-Ponceau-acid-fuchsin (Sigma-Aldrich, St. Louis, MO, USA) staining solution was added dropwise for 10 min, followed by 2% glacial acetic acid and phosphomolybdic acid dropwise for washing and differentiation, respectively. A 2% aniline blue dye solution (Sigma-Aldrich, St. Louis, MO, USA) was used for the final staining. Masson staining images and observation of myocardial tissue collagen were taken under a light microscope (Leica Microsystems, Wetzlar, Germany).

## TUNEL staining

Paraffin tissue sections were deparaffinized to water after baking at 65 °C for 1 h and then washed three times in PBS. Sections were fixed with freshly configured 4% formalin solution, room temperature, for 60 min. After three PBS washes, 3% methanol solution containing 0.2% $H_2O_2$ and sections were incubated for 15 min. The TUNEL reaction (Thermo Fisher Scientific Inc., Waltham, MA, USA) was configured according to the kit instructions and subsequently dropped onto the sections in the dark and placed in a wet box for 1 h incubation at 37 °C. After three PBS washes, DAB was added dropwise to the sections, which were finally blocked with a digoxigenin antibody buffer solution before PBS washing, and the images were acquired under a fluorescence microscope (Nikon, Tokyo, Japan).

## Immunofluorescence staining

Mouse left ventricular anterior wall tissues were collected, embedded in paraffin, sectioned, deparaffinized and rehydrated, washed five times with PBS buffer solution and treated with

Triton-X100 for 15 min. After three PBS rinses, sections were incubated with LC3II/I antibody (1:200; Abcam, Cambridge, United Kingdom) at 4 °C for 15 h, followed by the addition of secondary antibody treatment. After PBS washing three times, DAPI was added dropwise to the sections to stain the nuclei in the dark condition. Finally, the sections were blocked with fluorescence discharge quencher solution, and the images were acquired under a fluorescence microscope (Nikon, Tokyo, Japan).

## Western blot assay

Myocardial tissues frozen and stored at −80 °C were taken and minced with scissors, to which cold RIPA tissue lysate was added to extract proteins. After determining the protein concentration, equal amounts of protein per group were resolved by 10% SDS-PAGE and transferred to a PVDF membrane (Millipore, Burlington, MA, USA), which was blocked by 5% skim milk for 1 h at room temperature. The PVDF membranes were subsequently incubated with primary antibodies against FOXO3 (1:1,000; ab109629), p-FOXO3 (1:1,000; ab47285), Mst1 (1:1,000; ab51134), Sirt3 (1:1,000; ab217319), AMPK (1:1,000; ab214425), p-AMPK (1:1,000; ab129081) and GAPDH (1:2,500; ab181602) from Abcam overnight at 4 °C. On the next day, PVDF membranes were washed in PBS with 0.1% Tween-20 and then incubated with a peroxidase-conjugated goat anti-rabbit IgG secondary antibody for 2 h at room temperature. Next, the ECL working solution was incubated with the membrane and molecular imaging was performed using a molecular imager. The resulting images were photo analyzed with ImageJ analysis software for the developed protein bands.

## Quantitative real time PCR (RT-qPCR)

In order to extract total RNA from exosome pellets, tissues, or cells, TRIzol reagent (Takara, Tokyo, Japan) was used in this experiment. The above RNA was used as a template for reverse transcription, and RNA was reversed to cDNA using a Prime Script RT reagent Kit (Takara, Tokyo, Japan). Importantly, the RT-qPCR parameters were set according to an ABI 7900HT RT-PCR system settings (Thermo Fisher Scientific, Waltham, MA, USA) using the SYBR Premix Ex Taq II program. miR-200a-3p forward 5′-GGCTAACACTGTCTGGTAA CGATG-3′ and reverse 5′-GTG CAG GGT CCG AGG T-3′; U6 forward 5′-CAA ATT CGT GAA GCG TTCC ATA T-3′ and reverse 5′-GCT TCA CGA ATT TGC GTG TCA TCC TTG C-3′. The reaction system was 20 µL in volume: 2 µL of cDNA, 0.8 µL of up and downstream primers each, 10 µL of SYBR Green qPCR Master Mix reagent, ddH$_2$O added to 20 µL. Amplification conditions: 95 °C for 10 s followed by 40 cycles at 95 °C for 5 s, 60 °C for 31 s, and 72 °C for 15 s. Finally, the relative gene levels were expressed using the $2^{-\Delta\Delta CT}$ method.

## Luciferase gene reporter assay

FOXO3 wild-type and mutant (FOXO3-wt/mut) primers were designed at the predicted binding sites of the FOXO3 3′-UTR to miR-200a-3p and further synthesized by RT-qPCR. Subsequently, the above FOXO3 wild-type fragment of interest and mutant fragment were ligated separately with pMIR-REPORTTM vector using T4 ligase. The luciferase activity

was assessed using a Dual-Luciferase® Reporter Assay System (Promega, Madison, WI, USA). Using Lipofectamine 2000, HEK293T cells (BeNa Culture Collection, Shanghai, China) were transfected with miR-200a-3p mimics/NC and FOXO3-WT/MUT vector plasmids at room temperature. Then, the above mixture was detected luciferase activity.

## ELISA

The content of interleukin (IL)-6, IL-1β, TNF-α and ICAM-1 in the serum of mice was detected with IL-6, IL-1β, TNF-α and ICAM-1 ELISA kits (USCN, Wuhan, Hubei, China) according to the manufacturer's protocols.

## Statistical analysis

All the measurement data were statistically and analyzed using SPSS 22.0 software (SPSS, Inc., Chicago, IL, USA), and comparisons among multiple groups were performed by one-way ANOVA followed by Tukey's *post hoc* test, while the T-test was used for pairwise comparisons. The measurement data were expressed as mean ± standard deviation (SD). $P < 0.05$ was considered as statistically significant difference.

# RESULTS

## miR-200a-3p improved cardiac function in DCM mice

To investigate the biological function of miR-200a-3p in DCM *in vivo*, rAAV-miR-200a-3p or a corresponding negative control (rAAV-SC) were injected in mice. At the end of the animal experiment, the cardiac tissue proteins of mice in each group were taken for RT-qPCR analysis. The results showed that miR-200a-3p expression in the myocardium of DCM and DCM+rAAV-SC groups were substantially lower than that of normal mice in the control group (Fig. 1A). The rAAV-miR-200a-3p intervention could memorably increase the expression level of miR-200a-3p in the heart of DCM mice (Fig. 1A). To assess cardiac function in mice, echocardiography was performed. The ultrasound results showed that the LVEF and LVFS of DCM and DCM+rAAV-SC groups were significantly lower than those of control mice (Figs. 1B and 1C), suggesting that DCM mice developed LV systolic dysfunction at the end of the experiment. In addition, we further found that the LVEDD and LVESD of DCM and DCM+rAAV-SC groups were significantly increased compared with control mice (Figs. 1D and 1E), suggesting that DCM mice also developed LV diastolic dysfunction at the end of the experiment. Injection of rAAV-miR-200a-3p, the DCM+rAAV-miR-200a-3p group exhibited a significant increase in LVEF and LVFS (Figs. 1B and 1C), whereas LVEDD and LVESD showed a significant decrease (Figs. 1D and 1E), suggesting that upregulating miR-200a-3p expression can ameliorated diabetes induced-cardiac dysfunction in mice.

## miR-200a-3p relieved myocardial injury and cardiac fibrosis in DCM mice

To assess the myocardial injury and degree of cardiac fibrosis in mice, we performed HE staining and Masson staining on mouse heart sections. HE staining results showed that mice in the DCM and DCM+rAAV-SC groups developed obvious myocardial injury

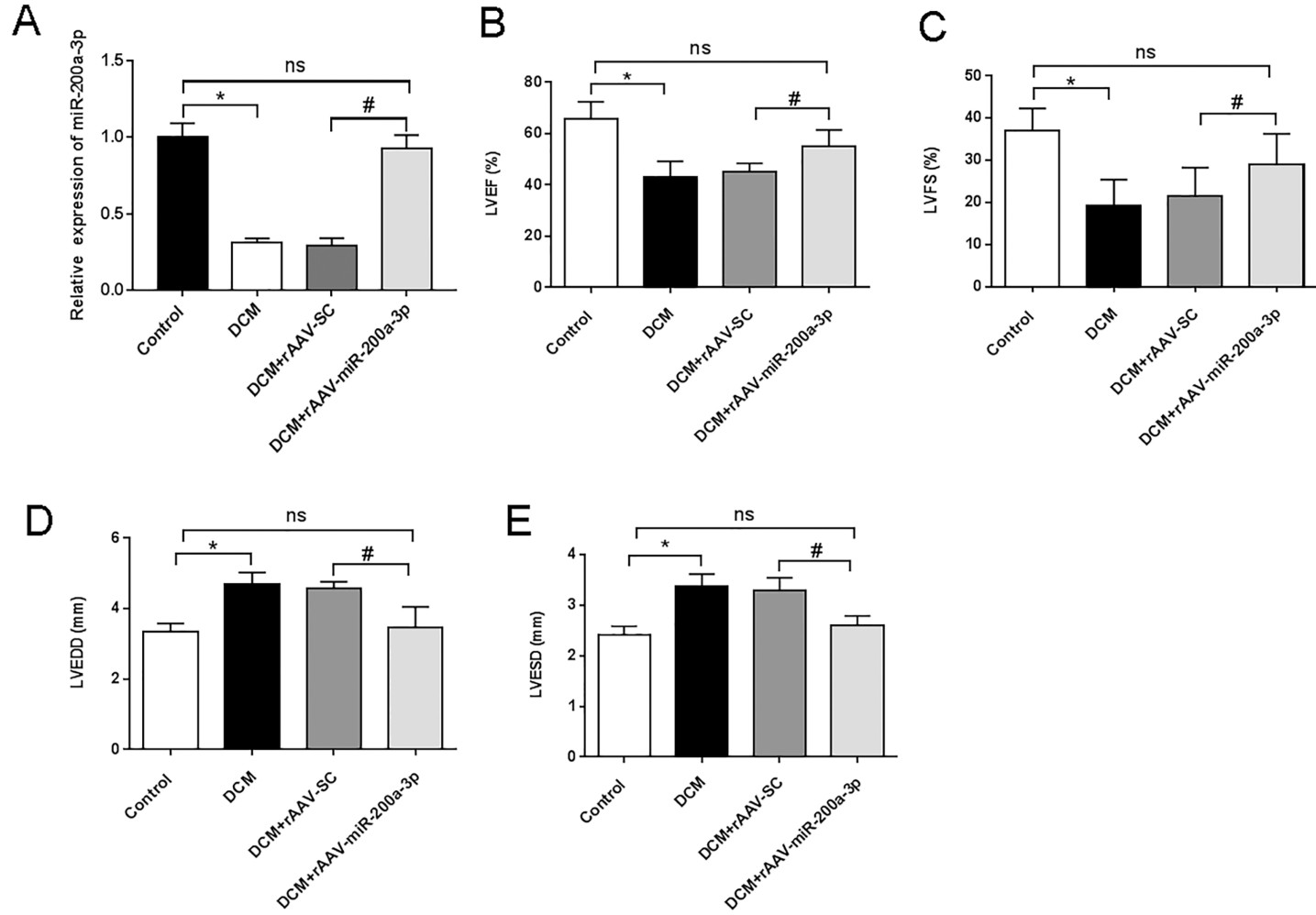

**Figure 1 miR-200a-3p reduced diabetes induced-cardiac dysfunction.** Control and DCM mice were injected with the corresponding rAAVs at 8 weeks of age and then sacrificed at 28 weeks of age ($n = 10$). (A) RT-qPCR measurement results of miR-200a-3p in mice ($n = 10$); (B and C) changes in LVEF and LVFS in mice ($n = 10$); (D and E) changes in LVEDD and LVESD in mice ($n = 10$). The experiments were performed at least five times. Data are expresses as SD. $^*P < 0.05$. *vs.* Control; $^#P < 0.05$ *vs.* DCM+rAAV-SC.

compared with the control group, while the mice in the DCM+rAAV-miR-200a-3p group received obvious alleviation of myocardial injury compared with the mice in the DCM+rAAV-SC group (Fig. 2A). Masson staining results proved that the deposition level of collagen fibers in the hearts of DCM mice was significantly higher than that in normal mice, and the injection of rAAV-SC did not affect the deposition level of collagen fibers, while overexpression of miR-200a-3p expression in the hearts of DCM mice decreased the interstitial collagen deposition caused by DCM (Fig. 2B). In addition, the levels of CK, CK-MB and LDH in the heart tissue of DCM mice were significantly higher than those of normal mice. Increasing the expression of miR-200a-3p in DCM mice could effectively reduce the production of CK, CK-MB and LDH (Figs. 2C–2E).

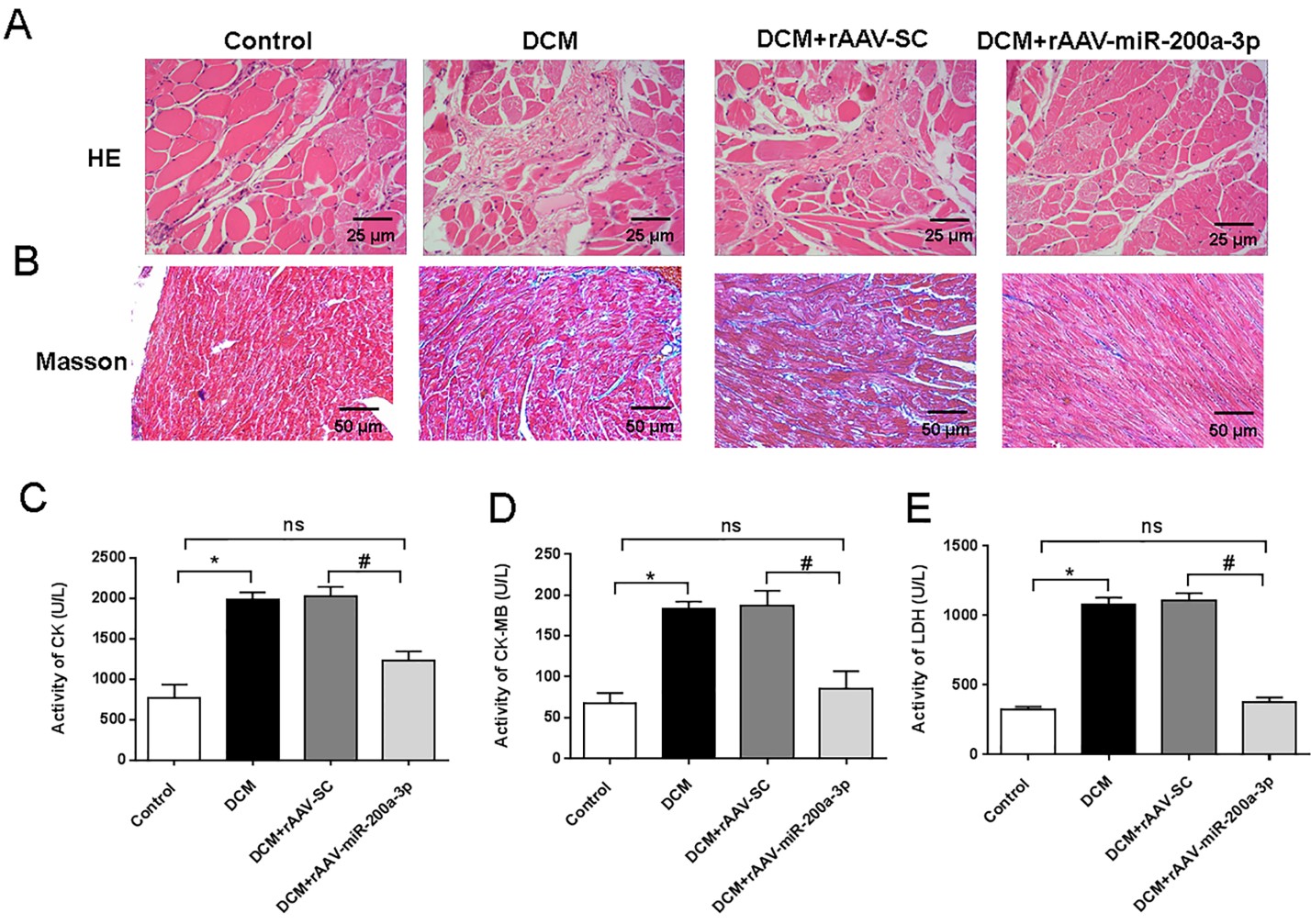

**Figure 2 miR-200a-3p reduced diabetes induced-myocardial injury and fibrosis.** Control and DCM mice were injected with the corresponding rAAVs at 8 weeks of age and then sacrificed at 28 weeks of age ($n = 10$). (A) HE staining of mouse myocardial tissues; (B) Masson staining of mouse myocardial tissues; (C–E) changes in CK, CK-MB and LDH production in mice ($n = 10$). The experiments were performed at least five times. Data are expresses as SD. $^*P < 0.05$ *vs*. Control; $^\#P < 0.05$ *vs*. DCM+rAAV-SC.

## miR-200a-3p suppressed the secretion of inflammatory factors in DCM mice

In view of the important role of inflammatory response in the occurrence and development of diabetes, we further explored the regulatory effect of miR-200a-3p on the inflammatory response during DCM. We assessed proinflammatory cytokines (TNF-α, IL-1β, IL-6 and ICAM-1) production. We collected the serum of DCM mice and found that proinflammatory cytokine levels were all prominently rose in DCM and DCM+rAAV-SC groups. Subsequently, our results revealed that the DCM+rAAV-miR-200a-3p group showed potent suppression of proinflammatory cytokine production in comparison to expression in the DCM+rAAV-SC group (Figs. 3A–3D).

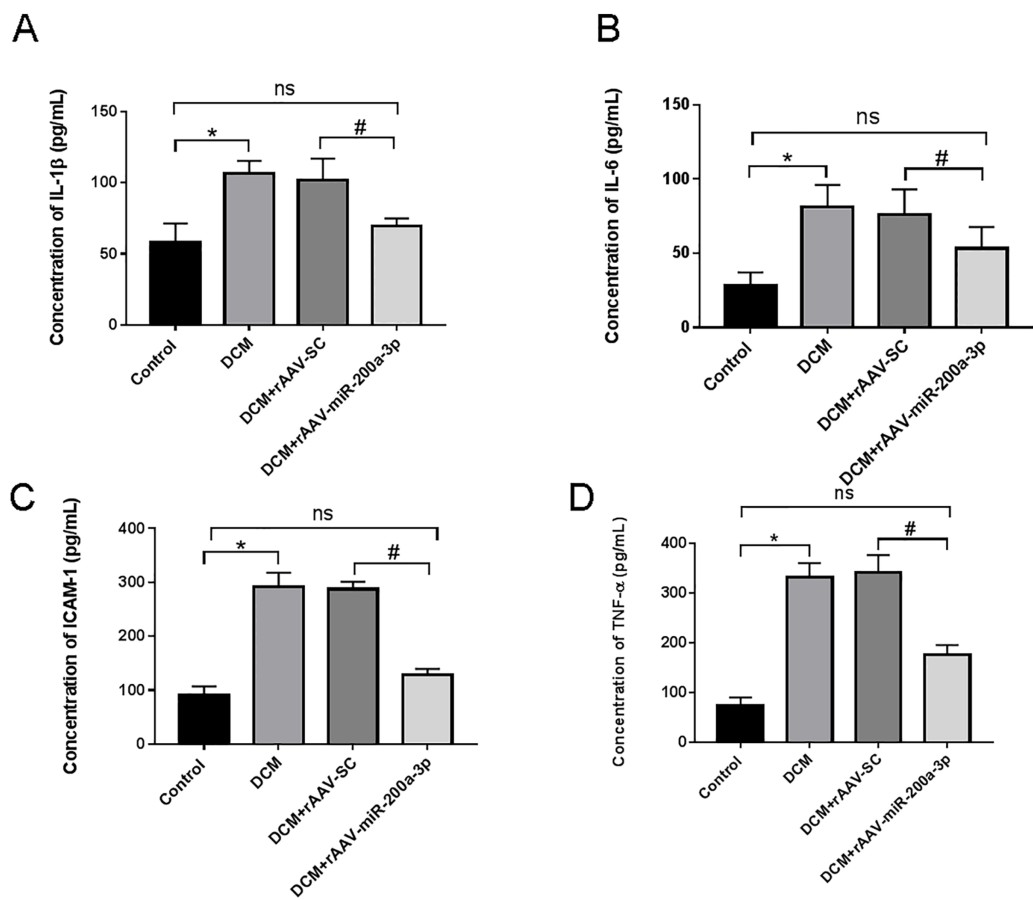

**Figure 3** **miR-200a-3p reduced diabetes induced-proinflammatory cytokine expression.** Control and DCM mice were injected with the corresponding rAAVs at 8 weeks of age and then sacrificed at 28 weeks of age ($n$ = 10). Proinflammatory cytokine levels, including IL-1β, IL-6, ICAM-1 and TNF-α, was examined by ELISA. (A–D) Changes in IL-1β, IL-6, ICAM-1 and TNF-α production in mice ($n$ = 10). The experiments were performed at least five times. Data are expresses as SD. [*]$P$ < 0.05 $vs.$ Control; [#]$P$ < 0.05 $vs.$ DCM+rAAV-SC.     

## miR-200a-3p promoted autophagy and prevented apoptosis of myocardial cells in diabetes mice

Typically, in high-fat diet induced type 2 diabetes, the occurrence of autophagy plays an important role in regulating cardiomyocyte survival. Therefore, we detected the occurrence of autophagy by measuring the expression of autophagy related proteins in myocardial tissues using Western blotting. We found that, at the protein level, the expression of Beclin-1 and LC3-II/I was significantly decreased in the hearts of mice in both DCM and DCM+rAAV-SC mice groups compared with that in control mice (Figs. 4A and 4B). In addition, the relative fluorescence intensity of immunofluorescence staining of LC3-II/I protein was dramatically decreased in DCM and DCM+rAAV-SC groups compared with the control group (Fig. 4C). The above results suggested that autophagic response was attenuated in the myocardium of diabetic mice. However, the expression of LC3-II/I and Beclin-1 as well as LC3-II/I protein fluorescence intensity in the myocardium of mice in DCM+rAAV-miR-200a-3p group was increased compared with

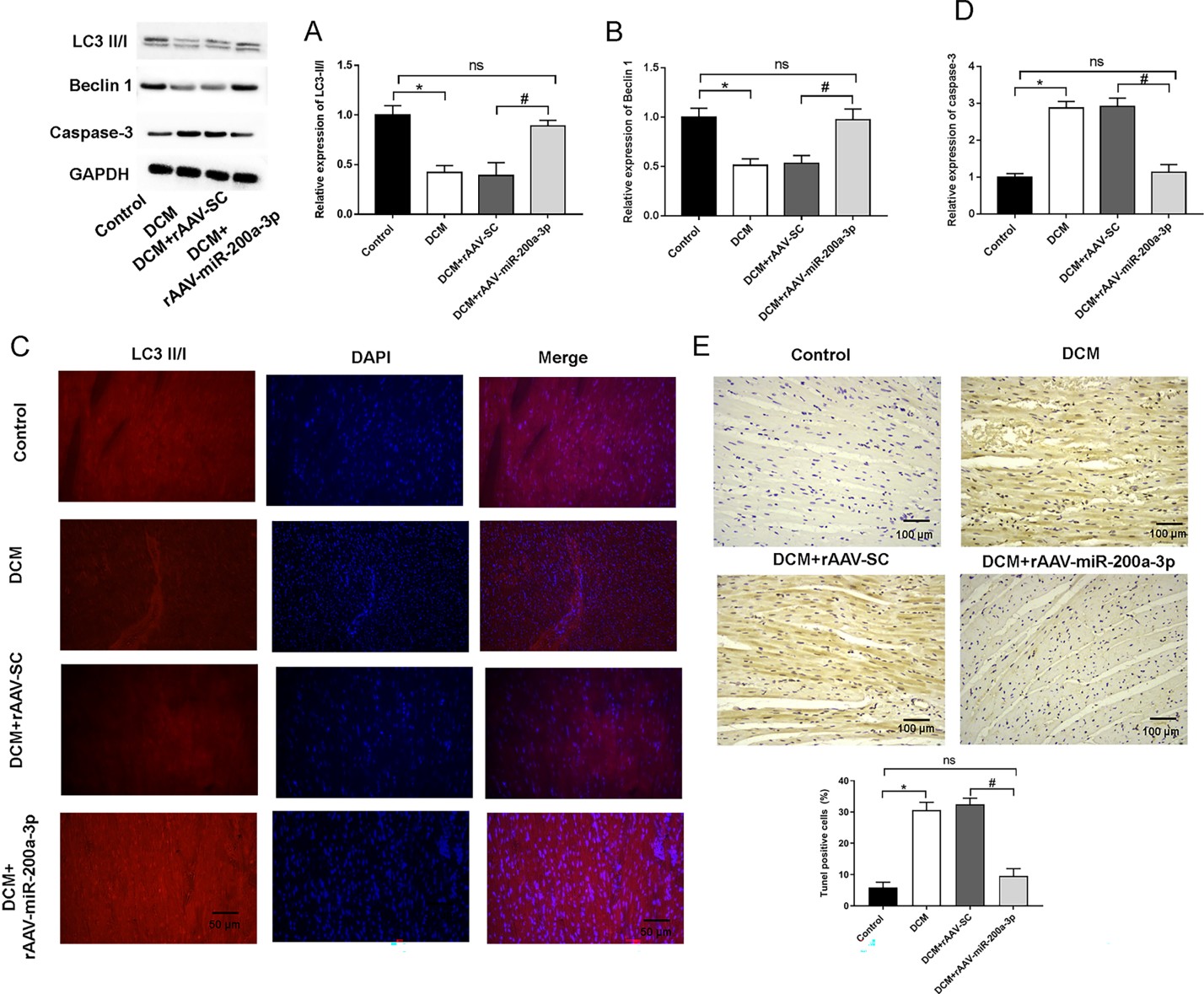

**Figure 4 miR-200a-3p affected diabetes induced-autophagy and apoptosis.** Control and DCM mice were injected with the corresponding rAAVs at 8 weeks of age and then sacrificed at 28 weeks of age ($n = 10$). (A and B) Western blotting measurement results of LC3-II/I and Beclin-1 in mice ($n = 10$); (C) immunofluorescence staining measurement results of LC3-II/I in mice ($n = 10$); (D) TUNEL staining measurement results of cell apoptosis in mice ($n = 10$); (E) Western blotting measurement results of caspase-3 in mice ($n = 10$); The experiments were performed at least five times. Data are expresses as SD. $^*P < 0.05$ *vs.* Control; $^\#P < 0.05$ *vs.* DCM+rAAV-SC.

that in DCM+rAAV-SC group of mice (Figs. 4A–4C). We further explored the role of miR-200a-3p in cardiomyocyte apoptosis in DCM by TUNEL staining and western blotting experiments of apoptosis key protein (caspase-3). The results showed that mice in the DCM and DCM+rAAV-SC groups exhibited significantly higher levels of apoptosis and caspase-3 expression than the control group, whereas those in the DCM+rAAV-miR-200a-3p group exhibited significantly lower levels of apoptosis and caspase-3 expression than those in the DCM+rAAV-SC group (Figs. 4D and 4E).

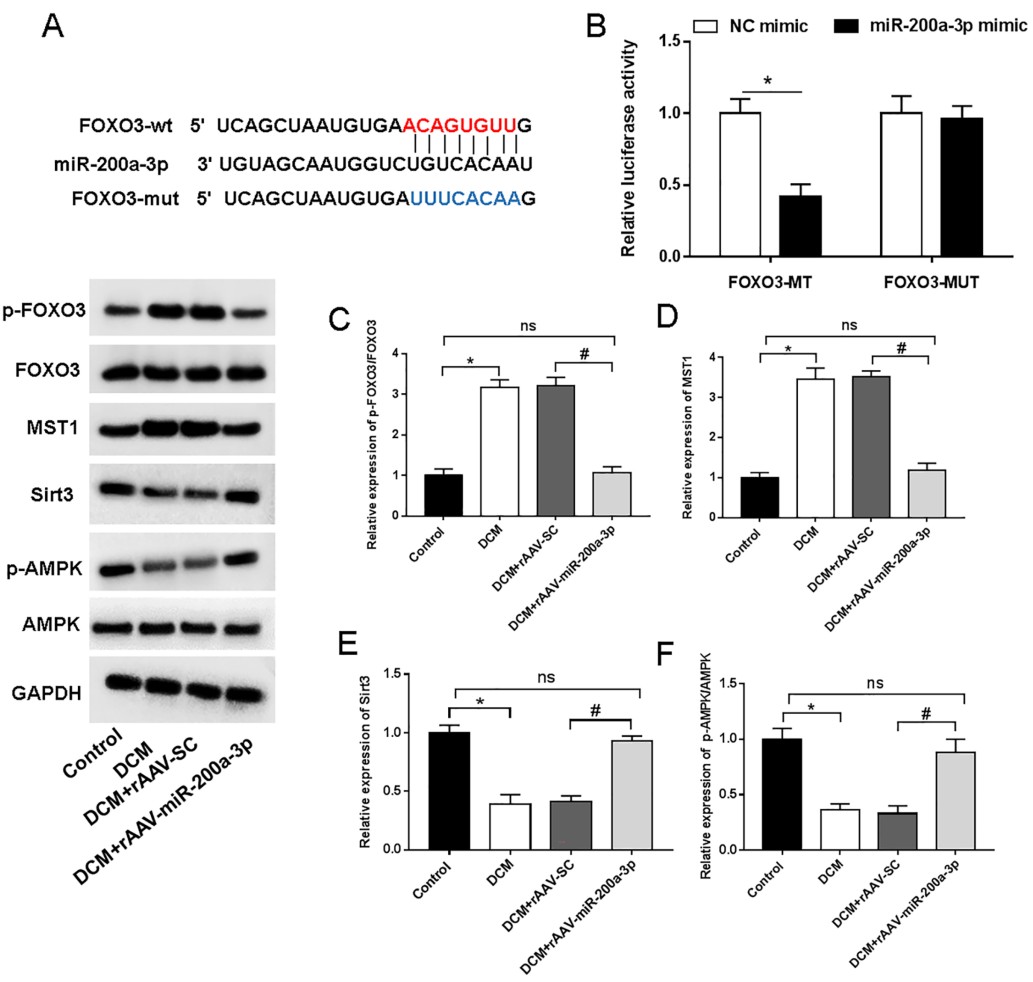

**Figure 5 miR-200a-3p reduced diabetes induced-myocardial injury through FOXO3/Mst1/Sirt3/ AMPK pathway.** Control and DCM mice were injected with the corresponding rAAVs at 8 weeks of age and then sacrificed at 28 weeks of age ($n = 10$). HEK293T cells were transfected with miR-200a-3p and NC mimic. (A) miR-200a-3p targeting sits of FOXO3 prediction analysis. (B) Dual luciferase reporter gene assay measurement results of luciferase activity; (C–F) R Western blotting measurement results of FOXO3, Mst1, Sirt3 and p-AMPK in mice ($n = 10$). The experiments were performed at least five times. Data are expresses as SD. $^{*}P < 0.05$ *vs*. Control; $^{#}P < 0.05$ *vs*. rAAV-SC.

## miR-200a-3p contributed to myocardial injury in diabetic mice through FOXO3/Mst1/Sirt3/AMPK pathway

We then aimed to examine the underlying mechanism of miR-200a-3p in DCM. StarBase (http://starbase.sysu.edu.cn/) demonstrated the targeting sites of miR-200a-3p in the FOXO3 3′-UTR (Fig. 5A). Furthermore, miR-200a-3p elevation substantially decreased the luciferase activity of FOXO3-wt, but FOXO3-mut abolished this effect (Fig. 5B). A previous study has suggested that FOXO3 acts on Mst1 and several other downstream factors to affect cell survival and cell function (*Xiong et al., 2020*). To explore the underlying mechanism of miR-200a-3p in diabetic myocardial injury, we examined the effects of rAAV-miR-200a-3p intervention on the FOXO3/Mst1/Sirt3/AMPK pathway.

Analysis of the Western blotting results revealed that db/db mice showed a significant increase in the levels of FOXO3 phosphorylation and Mst1 and decrease in the levels of Sirt3 and AMPK phosphorylation, whereas increasing of miR-200a-3p expression reversed the changes in the above indicators (Figs. 5C–5F).

## DISCUSSION

In this study, db/db mice combined with high-fat diet was used to induce DCM mouse model, and we observed that the expression of miR-200a-3p in myocardial tissues was decreased in DCM-induced myocardial injury. Furthermore, DCM mice developed LV dysfunction and cardiac fibrosis. Increasing miR-200a-3p expression levels improved LV function and alleviated cardiac fibrosis in DCM mice. According to a large number of substantial studies, it is well established that chronic low-grade inflammation is prevalent in diabetic patients in addition to disorders of glucose and lipid metabolism and abnormal subcellular composition (*Gu et al., 2020*). High glucose stimulation not only activates inflammasomes such as NLRP3 and AIM2, but also enhances TLR4 mediated inflammatory cascades and causes increased release of proinflammatory and proinflammatory factors such as IL-6 and TNF-α (*Xie et al., 2020*; *Wang et al., 2019*). Studies have shown that inhibition of the inflammatory response is beneficial for improving cardiac function in DCM (*Guo et al., 2016*; *Jin et al., 2021*). Here, we found that the levels of inflammatory factors were increased in the serum of mice under diabetic condition, and after overexpression of miR-200a-3p, the inflammatory response and myocardial injury caused by diabetes were significantly alleviated.

There is increasing evidence that miR-200a-3p is involved in a variety of diabetes complications. High levels of miR-200a-3p are found in the renal cortex resulting from aldose reductase deficiency, and inhibition of its expression downregulated Nrf2 and increases Keap1, TGF-β1/2, thereby exacerbating diabetes induced proteinuria production as well as renal injury and fibrosis, ultimately advancing the progression of diabetic retinopathy (*Wei et al., 2014*). In diabetic retinopathy (DR), miR-200a-3p was significantly decreased retina tissues of DR rats, and enforced of miR-200a-3p ameliorated retinal neovascularization and inflammation of DR rats (*Xue et al., 2020*). *Sun & Zhang (2019)* reported that myocardial infarction mice model and the hypoxia-induced myocardial cell with downregulation of miR-200a-3p, and miR-200a-3p was a potential candidate biomarker. All above studies suggested that miR-200a-3p were closely related to the occurrence and development of DCM.

DCM pathogenesis is complex and pathophysiologically characterized by myocardial fibrosis, limited cardiac motor function, and heart failure. In recent years, autophagy has become a research hotspot in the mechanism of DCM. Autophagy is a conserved biological behavior whose main process, the reuse after the degradation of intracellular components, plays a key role in maintaining energy balance and cell survival in response to nutrient/ energy stress (*Burman & Ktistakis, 2010*; *Levine & Klionsky, 2004*). Normally, sustained low-level autophagy in the heart is a protective mechanism against cellular stress (*Wu et al., 2017*). Disruption of cardiomyocyte autophagy, especially impaired autophagic flux, plays an important role in heart failure, hypertrophic cardiomyopathy, dilated

cardiomyopathy, aging, myocardial ischemia/reperfusion injury, and DCM (*Morales et al., 2020*). Therefore, autophagy has emerged as an important target for the prevention of DCM. Apoptosis and autophagy are universal phenomena in the occurrence and development of DCM. At present, there are increasing reports on the interaction between autophagy and apoptosis in DCM. *He et al. (2013)* found that hyperglycemia induced-cardiomyocyte apoptosis and inhibited autophagy was a cellular energy sensor that regulated apoptosis by promoting the interaction between Beclin1 and Bcl-2. *Ni et al. (2020)* revealed that high expression of miR-34a in the heart of diabetic mice, which inhibited autophagy, lead to cardiac dysfunction in mice. *Chen & Zhang (2021)* uncovered that GAS5 expression level was decreased in the myocardium of DCM model rats, and GAS5 improved the deteriorated cardiac function of DCM rats by downregulating miR-221-3p, upregulating p27, enhancing autophagy. Recent studies on the involvement of miR-200a in the regulation of autophagy have attracted extensive attention. *Guo et al. (2016)* demonstrated that in patients with anaplastic thyroid carcinoma (ATC), the level of miR-200a was significantly decreased, and miR-200a upregulation enhanced autophagy, thus promoting ATC cell apoptosis (*Miao et al., 2020*). In liver diseases, all-trans retinoic acid (ATRA) promoted the occurrence of autophagy in hepatic progenitor cells (HPCs) to induce hepatic differentiation of HPCs by upregulating miR-200a-3p (*Hu et al., 2019*). In current study, we found another important mechanism of cardiac autophagy in diabetes: miR-200a-3p. The downregulation of miR-200a-3p was the main reason for the inhibition of autophagy and the increase of apoptosis in mice. Overexpression of miR-200a-3p could, respectively, increase or decrease the autophagy and apoptosis of myocardial cells in m db/db mice.

Abnormalities in autophagy will contribute to abnormal cardiac function in diabetes. FOXO3 is a key protein, which is a transcription factor that can bind to the autophagy related protein ATG7 to activate its transcription (*Yu et al., 2019*). In our study, we demonstrated that FOXO3 was a target gene of miR-200a-3p. Therefore, downregulation of miR-200a-3p increased FOXO3 expression and inhibited autophagy, leading to the development of myocardial injury and fibrosis. In addition, FOXO3 has also recently been found to be a transcription factor for Mst1 and can positively regulate its expression in DCM (*Xiong et al., 2020*). Melatonin alleviated cardiac remodeling and dysfunction in DCM by upregulating autophagy and limiting apoptosis *via* blocking Mst1/Sitr3 signaling (*Zhang et al., 2017*). In addition, Sitr3 positively regulated AMPK activation, and activated AMPK was protective in diabetic hearts (*Xu et al., 2018*). In this study, we found that miR-200a-3p expression was significantly decreased in DCM and that the altered activity of the FOXO3/Mst1/Sitr3/AMPK pathway was mediated by miR-200a-3p expression.

## CONCLUSIONS

In summary, we show for the first time through animal studies that miR-200a-3p expression is decreased in an experimental DCM model and that miR-200a-3p upregulation can alleviate apoptosis by downregulating FOXO3, inactivating Mst1 transcription, promoting Sitr3 expression and AMPK phosphorylation levels, activating autophagy, and subsequently delaying the progression of diabetic cardiomyopathy, thus

providing a new idea and theoretical basis for the search of therapeutic approaches for the prevention and treatment of diabetic cardiomyopathy. This study also has some deficiencies, and lacks in-depth mechanistic exploration of the underlying regulatory mechanisms of aberrant miR-200a-3p expression and the signaling pathways and biological functions that miR-200a-3p participate in DCM.

### Funding
This research was supported by the Natural Science Foundation of Shaanxi Provincial (2022JM-540). The funders had no role in study design, data collection and analysis, decision to publish, or preparation of the manuscript.

### Grant Disclosures
The following grant information was disclosed by the authors:
Natural Science Foundation of Shaanxi Provincial: 2022JM-540.

### Competing Interests
The authors declare that they have no competing interests.

### Author Contributions
- Penghua You conceived and designed the experiments, analyzed the data, authored or reviewed drafts of the article, and approved the final draft.
- Haichao Chen performed the experiments, prepared figures and/or tables, and approved the final draft.
- Wenqi Han performed the experiments, prepared figures and/or tables, and approved the final draft.
- Jizhao Deng conceived and designed the experiments, analyzed the data, authored or reviewed drafts of the article, and approved the final draft.

### Animal Ethics
The following information was supplied relating to ethical approvals (*i.e.*, approving body and any reference numbers):
All experimental procedures and protocols were approved by the Ethics Committee of Shaanxi Provincial People's Hospital.

### Data Availability
The raw data are available in the Supplemental Files.

### Supplemental Information
Supplemental information for this article can be found online at http://dx.doi.org/10.7717/peerj.15840#supplemental-information.

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
