# Peer review of "miR-200a-3p overexpression alleviates diabetic cardiomyopathy injury in mice by regulating autophagy through the FOXO3/Mst1/Sirt3/AMPK axis"

_PeerJ, doi:10.7717/peerj.15840_

## Round 0.1 · original submission · Minor Revisions

A point to point response letter and language editing service.

Reviewer 1 ·

Basic reporting

This study, You et al, investigated the role and mechanism of miR-200a-3p in diabetic cardiomyopathy (DCM). Although protective effects of miR-200a-3p are significant findings, this study is still descriptive and falls short in mechanistic advancement. Some major issues need to be revised.
A. This study shows that miR-200a-3p inhibitor aggravates diabetic cardiomyopathy injury. However, the data showed that miR-200a-3p overexpression could alleviate diabetic cardiomyopathy injury, and miR-200a-3p inhibitors are not mentioned. Therefore, it is suggested in the title to mention miR-200a-3p overexpression instead to miR-200a-3p inhibitor.
B. In the introduction, it is suggested to mention the prevalence of diabetic cardiomyopathy.

Experimental design

C. Please mention all forward and reverse primer sequences used during PCR.
D. Mice in the model group were fed a high-fat diet. Please clarify the specific formula of the high-fat diet.
E. Please specify the percentage isoflurane used for the induction and maintenance of anaesthesia.
F. Please clarify, were all mice euthanised by removal of the heart under anaesthesia? And also specify what method of anaesthesia was used for this.

Validity of the findings

.

Additional comments

.

Reviewer 2 ·

Basic reporting

This article draws attention to miR-200a-3p as a targeted therapeutic approach for diabetic cardiomyopathy. However, there are several points that need to be clarified as follows:
1) The author indicates that the DCM model was established in db/db mice. But, the db/db mice have shown symptoms of myocardial injury in diabetes, and no further construction of DCM model is required. So, this sentence needs reframing for better readability.
2) There is too much description of other miRNAs in the section of introduction. The miR-200a-3p should be described in detail in the introduction.
3) The author need provide more information about the miR-200a-3p mimic/inhibitor and negative control. And, are these scramble sequences? Moreover, how were the doses of miR-200a-3p inhibitor/negative control selected?
4) In general, the codes of the kits used should be reported in the materials & methods section. The brand and model of reagents and experimental instruments in the methodology one by one.
5) What method is used for post hoc testing after analysis of ANOVA?
6) The author need provide a reference for this sentence ' Previous studies have suggested that FOXO3 acts on Mst1 and several other downstream factors to affect cell survival and cell function'.
7) Lack of in-depth exploration mechanism is a major problem in this study. The main defects of this study should be clarified in the discussion section.
8) The illustrations of all the figures should be described in more detail and need to indicate the number of biological repetitions for each experiment.

Experimental design

no comment

Validity of the findings

no comment

Additional comments

no comment

---

## Round 0.2 · Minor Revisions

Please attend to the following feedback from the Section Editors:

The manuscript needs better data presentation in order to justify the title. Also, reference numbers for the antibodies are not disclosed (impossible to properly reproduce the experiments). Please ensure the authors meet the PeerJ policy for reporting antibodies at https://peerj.com/about/policies-and-procedures/#data-materials-sharing - and ideally that you provide RRIDs (Research Resource IDentifiers) if available.

There are also issues with the histopathology; for example, you named "Masson staining" instead of Masson's trichrome staining and you say "Masson Ponceau acid fuchsin when acid fuchsin is the same as Masson...).

Fig 4C is not big enough to distinguish anything. The apoptotic index does not seem to match histology slides.

You must provide the original blots with the protein ladder included as blots and graphs do not seem to match well.

---

## Round 0.3 · accepted · Accept

The authors have addressed the previous comments.